# High-Fat Diets with Differential Fatty Acids Induce Obesity and Perturb Gut Microbiota in Honey Bee

**DOI:** 10.3390/ijms22020834

**Published:** 2021-01-15

**Authors:** Xiaofei Wang, Zhaopeng Zhong, Xiangyin Chen, Ziyun Hong, Weimin Lin, Xiaohuan Mu, Xiaosong Hu, Hao Zheng

**Affiliations:** College of Food Science and Nutritional Engineering, China Agricultural University, Beijing 510642, China; xiaofei.wang@cau.edu.cn (X.W.); zhongzhaopeng@cau.edu.cn (Z.Z.); chenxiangyin@cau.edu.cn (X.C.); redziyun@icloud.com (Z.H.); 2018306100224@cau.edu.cn (W.L.); xiaohuan.mu@cau.edu.cn (X.M.); huxiaos@263.net (X.H.)

**Keywords:** gut microbiota, dietary fat, fat accumulation, lipid metabolism, honey bee

## Abstract

HFD (high-fat diet) induces obesity and metabolic disorders, which is associated with the alteration in gut microbiota profiles. However, the underlying molecular mechanisms of the processes are poorly understood. In this study, we used the simple model organism honey bee to explore how different amounts and types of dietary fats affect the host metabolism and the gut microbiota. Excess dietary fat, especially palm oil, elicited higher weight gain, lower survival rates, hyperglycemic, and fat accumulation in honey bees. However, microbiota-free honey bees reared on high-fat diets did not significantly change their phenotypes. Different fatty acid compositions in palm and soybean oil altered the lipid profiles of the honey bee body. Remarkably, dietary fats regulated lipid metabolism and immune-related gene expression at the transcriptional level. Gene set enrichment analysis showed that biological processes, including transcription factors, insulin secretion, and Toll and Imd signaling pathways, were significantly different in the gut of bees on different dietary fats. Moreover, a high-fat diet increased the relative abundance of *Gilliamella*, while the level of *Bartonella* was significantly decreased in palm oil groups. This study establishes a novel honey bee model of studying the crosstalk between dietary fat, gut microbiota, and host metabolism.

## 1. Introduction

Dietary fats vary greatly in fatty acid composition and growing evidence demonstrates the heterogeneity in the health effects of specific fatty acids as well as their food sources [1,2]. Animal-derived fat includes lard, butter, and milk fat, while plant-derived fat includes soybean oil, corn oil, peanut oil, olive oil, rapeseed oil, palm oil, etc. Soybean oil contains a significant proportion of linoleic acid (51%) and oleic acid (26%), while palm oil has a high percentage of oleic acid (44%) and palmitic acid (37%). Palmitic acid is a saturated fatty acid (SFA) and the intake of palm oil may give rise to elevated total cholesterol and low-density lipoprotein cholesterol levels [3]. Additionally, the amount of dietary fat can affect host health. Excessive soybean oil intake causes obesity and inflammation in mice, which is linked to the damage to metabolic regulatory pathways, such as insulin resistance and fatty acid oxidation [4].

The human intestinal tract harbors a vast ensemble of microbiota, which play an important role in food digestion and host metabolism, including the ability to provide supplementary nutrients to the host and participate in necessary metabolic capabilities [5]. Recent studies indicate that dietary fats influence the gut microbiota composition, which may affect host metabolic health and result in epidemic increases in obesity-associated diseases [6,7]. A high-fat diet (HFD) causes an increase in the relative abundance of *Firmicutes* and a reduction in *Bacteroidetes* [8]. In addition to the fat amount, different types of fat can lead to dysbiosis of microbiota. SFA diets reduce microbial diversity and richness [9]. In contrast, intake of unsaturated fatty acids (UFA), such as n-3 polyunsaturated fatty acids (PUFA)-rich diets, exerts opposite effects, including increases in microbiota diversity and ratio of *Bacteroidetes* to *Firmicutes* [10,11]. Therefore, the study of biochemical mechanisms involved in host metabolism responses to specific dietary fat requires close attention to both gene expression at the transcriptional level and relative abundance at the gut microbiota level.

Given the complexity of the whole-body response to dietary fat changes, invertebrate model organisms can serve as useful tools to examine the interplay between genes, signal pathways, gut microbiota, and host metabolism [12]. In addition, regulatory pathways such as insulin signaling and TOR are highly conserved in invertebrates, as in mammals [13,14]. The honey bee gut possesses a simple and host-specific gut microbiota, which has served as a model system for the study of gut microbiota [15]. This gut community is dominated by five to nine taxa, which together account for >98% of bacterial 16S rRNA gene sequences. Typically, *Lactobacillus* Firm-5 is most abundant, followed by *Lactobacillus* Firm-4, *Bifidobacterium* spp., *Gilliamella apicola*, and *Snodgrassella alvi* [16]. Newly-emerged adult worker bees are almost free of bacteria, and gut microbiota is established within ~7 days after emergence through social interactions [15]. Thus, microbiota-free (MF) bees can be easily generated by pulling out the mature pupae with pigmented eyes but lacking movement from the capped brood and housing them in sterile plastic cup cages [16]. Using MF bees, it has been found that gut microbiota can promote honey bee health via the digestion of polysaccharides and the microbial metabolism of amino acids and short-chain fatty acids [17,18], while it is unknown if they participate in the metabolism of other dietary nutrients, such as lipids. Pollens are the main source of bee lipids, which are important for the synthesis of reserve fats and glycogen, and the membrane structure of cells [19,20]. The fatty acid types of pollens from different plants are basically consistent, including palmitic acid (C16:0), stearic acid (C18:0), oleic acid (C18:1), linoleic acid (C18:2), and linolenic acid (C18:3) [20]. Similarly, the main fatty acids in edible oils for humans are of the same type, while the relative composition differs [21]. It has been shown that sage and flax oil diets significantly improve hypopharyngeal gland sizes, as well as olfactory and tactile associative learning abilities compared to corn and sesame oil diets [22]. Supplementation of oleic acid and linoleic acid influences honey bee longevity, life-span, and head weight, as well as the midgut lipase activity [23]. Thus, the honey bee may provide a promising model to study the dietary fatty acid–gut microbe interactions.

In this study, we used the honey bee (*Apis mellifera*) to investigate the impact of different amounts and types of dietary fat on the host’s metabolism. Specifically, we used honey bees with a conventional gut microbiota (CV) and MF raised under axenic conditions in the lab to explore the roles of gut microbiota in fat absorption and digestion. We investigated the impact of dietary fats on gut microbiota and host metabolism in the honey bee. The influences on body weight gain, glucose and trehalose concentrations, lipid content, and composition were measured. Transcriptomics of the gut epithelial cells showed host lipid metabolism and immune-related pathways were upregulated with the soybean oil diet. Moreover, the gut microbiota composition and functions were changed according to 16S rRNA and phylogenetic investigation of communities by reconstruction of unobserved states (PICRUSt) analysis. This proved the feasibility of the honey bee as a model organism to study dietary fat and gut microbiota.

## 2. Results

To examine dietary-fat-induced effects on host metabolism, we fed honey bees palm or soybean oils at different concentrations. The normal diet contained sucrose as the carbohydrate source, casein amino acids as the protein source, together with 1% (*v*/*v*) of soybean (NDS) or palm oil (NDP). The HFD contained 5% or 10% (*v*/*v*) of the soybean (HFDS) or palm oil (HFDP). However, 10% of HFD is extremely toxic to the bees (Appendix A), thus we used 5% HFD for the establishment of the obesity model. MF and CV honey bees were obtained from pupae emerged under sterile conditions in the lab (Figure 1A).

### 2.1. Increased Dietary Fat Causes Metabolic Syndrome in Honey Bees

We first examined the dietary-fat-induced effects on the host’s metabolism. Honey bees on HFDP exhibited a higher percentage of body weight gain compared to NDP, regardless of the presence or absence of the gut microbiota (Figure 1B), which is probably due to the greater cumulative energy intake (Appendix A). In contrast, a HFDS caused a decreased body weight gain in honey bees with gut microbiota, while no significant change was observed for MF bees. Exposure to HFDP resulted in a reduction in survival rate compared with the NDP, while the presence of gut microbiota alleviated such effect (Figure 1C). However, in MF bees on a HFDS, the survival rate significantly increased, while it did not change for CV bees compared to those on a NDS. We next tested whether a HFD induced hyperglycemia in honey bees. HFDP significantly increased the level of hemolymph glucose in CV bees, compared to NDP. Moreover, for both NDP and HFDP, the glucose level in bee individuals with a gut microbiota was significantly higher than that of MF bees (Figure 1D), confirming the roles of gut microbiota in energy metabolism [24]. However, a soybean oil diet did not result in any hyperglycemia phenomenon in the honey bees. Additionally, the profile of trehalose content, a primary energy source in insect tissues and organs, was similar to that of glucose. These results indicate a specific adverse effect of HFD by palm oil on the honey bee’s homeostasis, which is also markedly regulated by the gut microbiota.

In both mammals and *Drosophila* model, an increased level of triglycerides (TAG) is a major risk factor for metabolic syndrome associated with obesity and hyperglycemia [25,26,27]. TAG is also the main form of stored fat in honey bees [28]. Thus, we determined the changes of TAG, total fatty acid (TFA), and the TAG/TFA ratio of the whole body induced by different dietary fats. We found an increase in the TAG content per body mass when bees were fed HFD of palm oil, both with and without the gut microbiota (Figure 2A). Although no significant differences were observed between HFDS and NDS, CV bees had a higher TAG concentration than MF bees. Since we measured the TFA of the whole body, which includes polar lipids, free fatty acids, and TAG, the ratio of TAG/TFA was calculated to indicate the adipose accumulation. The TFA content was not changed by either HFDS or HFDP (Figure 2B), however, both HFD diets elevated the TAG/TFA ratio in CV bees (Figure 2C). Interestingly, all groups of CV bees showed a higher TAG/TFA ratio, compared to bees with depleted gut community. Then, we investigated the accumulation of lipids in adipose tissue (fat body, FB), which is the major organ for lipid storage and functions as both adipose and liver in honey bees [29]. Using hematoxylin-eosin staining, we observed more lipid storage droplets with larger size in HFD-treated bees for both soybean and palm oil, compared to bees on a normal diet with 1% Lipid A as the sole fat source (Figure 2D), which aligns with the higher TAG level in bees on a HFD. These findings indicate that when fed a HFD, honey bees accumulate excess fat in adipose tissue, as seen in mammals and *Drosophila* [30].

Given that the palm and soybean oils elicited distinct aberrant phenotypes, we hypothesized that the deleterious effects are attributed to the dietary fatty acid composition. Palm oil that was supplemented into the bee diet contains higher levels of SFA (C16:0 and C18:0) and more monounsaturated fatty acids (MUFA; C18:1) than the soybean oil. Soybean oil is mainly constituted of PUFA (C18:2 and C18:3). Honey bees consuming palm oil possessed higher proportion of C18:1 in both TFA and TAG than those fed soybean oil (Figure 2E). Although soybean oil has less SFA, bees on the soybean oil diet showed a similar level of C16:0 and C18:0 in the body compared to the palm oil diet. The percentage of PUFA in soybean oil group was much higher than that of palm oil group, which is consistent with the relative abundance in the diet. Notably, the amount of fat intake and the gut microbiota had no striking effects on the fatty acid composition in the body. These confirm that the composition of different types of fatty acid in the diet altered the lipid profiles of the honey bee body, which may explain their effects on metabolic syndromes.

### 2.2. Transcriptional Changes in Lipid Metabolism and Immune-Related Pathway

To discover the biological pathways potentially contributing to the observed symptoms, we examined the transcriptional profiles of the gut epithelial cells. Here, we performed RNA-seq on CV bees fed different diets. Principal coordinate analysis (PCoA) showed that the dietary fat type has a striking effect on the transcriptomic profiles (Figure 3A). The samples of NDP and HFDP were not clearly separated, and the volcano plot showed no genes were differentially expressed between these two groups (Appendix A). There were 369 and 58 differentially expressed genes (DEGs) upregulated in HFDS and NDS, respectively (Appendix A). Specifically, genes related to Toll and Imd signaling pathway and fatty acid degradation were differentially expressed (Figure 3B). HFD of soybean oil suppressed the expression of genes encoding peptidoglycan recognition protein (PGRP and Pgrp-s2), poor Imd response upon knock-in (PIRK), membrane bound O-acyl transferase (MBOAT), and the antimicrobial peptide abaecin. Conversely, farnesyl diphosphate synthase (FPPS) and neurexin (NRXN) were significantly upregulated with the HFDS (Figure 3B). Gene set enrichment analysis showed that several KEGG pathways were significantly different in the gut of bees on HFDS. Pathways including transcription factors, insulin secretion, and bile secretion were upregulated in HFDS gut. Different sources of dietary fat also induced differential expression patterns of various genes. We found 3-hydroxybutyryl-CoA dehydrogenase (HADHA), 3-hydroxybutyryl-CoA dehydratase (HACD), and carnitine-palmitoyltransferase (CPT) were upregulated in bees fed soybean oil, compared to those on HFDP and NDP (Figure 3B). Expressions of acid-sensing ion channel (ASICN), cell division control protein (CDC45), Delta(3)-Delta(2)-enoyl-CoA isomerase (DCI), sphingomyelin phosphodiesterase (SMPD), and tumor necrosis factor alpha (TNF-α) were increased in palm-oil-fed bees. The enrichment analysis revealed that pathways involving Toll and Imd signaling pathway, unsaturated fatty acid biosynthesis, fatty acid degradation, and elongation were stimulated in bees on a NDS (Figure 3C). These findings indicated that dietary fats impact the transcriptional profiles of the gut tissue cells, and genes involved in the lipid metabolism and immune functions were primarily altered.

### 2.3. HFD Perturbs the Structure and Functional Profile of the Gut Microbiota

Gut microbiota dysbiosis is always associated with metabolic disorders induced by different dietary patterns [31]. To assess the effects of dietary fats on the composition of the microbiota, we analyzed the gut communities based on the V3–V4 region of 16S rRNA in ND- and HFD-treated bees. Although different dietary fats changed the gut composition, all core species of the bee gut were present with *Lactobacillus* Firm-5 as the most abundant member (Figure 4A) [18]. The microbial community richness was slightly increased in both HFD groups as measured by the Chao1 index (Figure 4B). The Shannon diversity index was higher in bees on a HFDS, compared to the NDS, while it was not altered by a HFD of palm oil (Figure 4C). Notably, the bee gut γ-Proteobacteria, *Gilliamella*, was significantly enriched upon HFD for both soybean and palm oil (Figure 4D). In bees fed palm oil, *Bartonella* was almost eradicated, indicating a divergent effect of dietary fat types on bee gut members (Figure 4E).

We further analyzed the shift of functional potential of the gut microbiota using the PICRUSt analysis. PCA analysis revealed distinct clusters among the four dietary groups, which indicates that the intake of dietary components results in characteristically different natures of the microbial metabolism (Figure 4F). Pathways involving carbohydrate metabolism, transcription, cellular processes and signaling, and metabolism of cofactors and vitamins were enhanced in both HFD groups compared with ND groups, while metabolism of terpenoids and polyketides was suppressed by the HFDs (Figure 4G). Bees on the soybean oil diet with higher PUFA content had a higher capacity of the microbial lipid metabolism. This may be because PUFAs are more difficult to transport and utilize than SFAs and MUFAs [32]. Furthermore, soybean oil increased the pathways related to xenobiotics metabolism, cell motility, cell growth and death, and membrane transport in the microbiota. In contrast, the palm oil diet upregulated pathways related to energy metabolism and signaling molecules. In summary, our results indicated that both the amount and type of dietary fats can cause significant changes to the structure and functional potential of the honey bee gut community.

## 3. Discussion

Honey bees rely on pollen as their main lipid source and normally the dietary fat concentration is ~1% [20]. Here, we used 5% and 10% HFD to generate obesity phenotypes. The 10% HFD induced a significantly shorter life span (Appendix A); this lethal effect of extraordinarily high levels of dietary fat is also observed in *Drosophila* [25]. However, 5% HFD gave stable and obvious detrimental effects. We found that honey bees fed a HFDP exhibited typical features of metabolic syndrome, including increased body weight gain and changes in glucose homeostasis, as found in *Drosophila* and mammalian models. Furthermore, the effects of HFDP and HFDS on adipose accumulation are profound, including elevated lipid levels, more lipid droplets, and larger droplet size. The HFD-induced elevation in TAG levels is an important marker for this collection of phenotypes, as high TAG levels are associated with disruption of lipid and glucose homeostasis, mitochondrial function, and other processes [33,34], all of which may contribute to metabolic disorders. Since the fatty acid composition is a key predictor of edible oil, we investigated the effects of both the soybean and palm oil in altering honey bee metabolism under HFD conditions. Palm oil contains a higher amount of SFA (C16:0 > 37%) and MUFA (C18:1 > 44%), while soybean oil is composed of higher levels of PUFA (C18:2 > 50%). Intriguingly, HFDS blocks the increased body weight gain, glucose and trehalose levels in the hemolymph of honey bees. These suggest that the relative abundance of fatty acids is a predisposition influencing the effects of the HFD phenotype on obesity and other metabolic disorders. Studies have shown that only high SFA diets confer obesity models, such as lard oil (SFA > 40%) for the establishment of obese rats, and coconut oil (SFA > 90%) is always employed in *Drosophila* model [30,34,35]. Alterations of the honey bee metabolic status associated with dietary fat source were accompanied by significant changes in the body lipid profiles, especially the abundance of oleic acid (C18:1) (Figure 2). Compared to palm oil, soybean oil diet lowered the survival rate, which suggests that oleic acid is more adapted to honey bees than linoleic acid (C18:2).

Metabolic regulation in honey bees shows important similarities to mammals. There is a high degree of conservation of many biological pathways: glucose transport and metabolism, lipid synthesis and storage, trehalose synthesis and stability, oxidative stress, insulin signaling pathway, and Toll and Imd signaling pathway. HFDS suppressed the expression of PGRPs, which promote gut immune homeostasis and extending of the lifespan in *D. melanogaster* [36]. Pgrp-s2 indeed plays a defensive role in honey bee, and is upregulated after disease challenge [13]. Furthermore, HFDS also inhibited the expression of PIRK, a protein that antagonizes PGRP signaling capacity and participates in the precise control of Imd pathway induction in the gut [37]. The expression levels of abaecin and def1, which encode antimicrobial peptides and are induced and controlled by Toll and Imd signaling pathways [38,39], decreased with the HFDS diet. In addition, MBOAT, which functions as a lysophospholipid acyltransferase on lipid droplet membrane, was downregulated in the HFDS. It has been linked to fibrosis and chronic hepatitis in humans, suggesting that HFD increases the fat accumulation and likelihood of adiposity-induced syndromes in honey bees [40]. The upregulated FPPS upon HFD is an enzyme important in the formation of farnesyl pyrophosphate, an intermediate in cholesterol synthesis [41]. FPPS is significantly upregulated in diabetic animals [42]. Thus, transcriptomic data revealed numerous DEGs related to Toll and Imd signaling pathway and lipid metabolism in a HFD, which parallels previous findings in *Drosophila* and mammalian models. Comparing bees on a SFA-rich palm oil diet and those on a soybean oil diet, the expression levels of CPT were decreased. CPT catalyzes the rate-limiting step for entry of long-chain fatty acyl CoA into mitochondria, and the SFA- and sugar-rich Western diet suppresses CPT in humans [43]. In the palm oil group, the expression level of TNF-α was upregulated, suggesting that higher SFA intake may contribute to intestinal permeability and inflammation [44]. In contrast, PUFA supplementation can decrease the levels of TNF-α expression in rat colon and has anti-inflammatory effects [11,45].

PICRUSt analysis predicted that different dietary fats can cause changes in the functional capacity of gut microbiota. HFD increased the relative abundance of *Gilliamella*, a dominant member belonging to γ-proteobacteria, compared to ND for both soybean oil and palm oil groups. In mice, a HFD can increase the relative abundance of γ-Proteobacteria (*Enterobacteriaceae*) and δ-Proteobacteria (*Bilophila*) [46]. The compositional shift was accompanied by a functional change, as a HFD prompted increased sucrose metabolism, membrane transport systems, and metabolism of cofactors and vitamins [31]. To incorporate and emulsify the excess dietary fat into mixed micelles, *Gilliamella* might further strengthen its carbohydrate metabolism and cofactor metabolic process to produce more HFD-specific metabolites as energy substrates for the formation of fat emulsion [47]. PUFAs are more difficult to combine with bile acids and are less effectively transported and utilized than SFAs and MUFAs. The PUFA-rich soybean oil induced a higher capacity of microbial lipid metabolism in the gut. Likewise, cecal bacteria of mice incubated with fish oil show higher abundance of lipid metabolism [35]. Intriguingly, *Bartonella* was enriched in the soybean oil groups. Winter bee gut microbiota are remarkably dominated by *Bartonella*, which may play an important role in modulating energy metabolism [48]. Altogether, the amount and the fatty acid structure of dietary fats can regulate the composition and functional profiles of the gut community, which is important to the lipid absorption and metabolism.

In summary, the discovery of HFD-induced obesity model in honey bees permits a detailed dissection of obesity phenotypes, from metabolic syndromes to transcriptional changes in host metabolism, and the functional changes of gut microbiota. In particular, we can attempt to understand the various contributions of dietary fat amounts and fatty acid profiles to the phenotypes associated with altered gut microbiota. Moreover, it is potential to recommend dietary fat intake through adjusting both amount and type, particularly by lowering the intake of SFA and increasing PUFAs, which is in line with dietary reference guidelines to promote human health.

## 4. Materials and Methods

### 4.1. Animal Experiments

MF and CV bees were obtained according to the protocol described by Powell et al. [49]. For the MF bees, late-stage pupae were removed manually from the brood frames of hives collected in Shunyi, Beijing, and were placed in sterile plastic bins. The pupae were allowed to emerge in a sterilized growth chamber at 35 °C. Fifteen to thirty newly emerged individuals were kept in an axenic cup-shaped hoarding cage with a removable base and ventilation holes. The bees were fed with sterilized sucrose syrup (0.5 M). Batch sterilizations were verified by plating suspensions on LB plates and incubating them at 37 °C overnight. The CV bees were obtained by feeding emerged bees with homogenates of freshly dissected hindguts of nurse bees from their hives of origin, in addition to their food. MF and CV bees were fed sucrose solution for 2 d. Before diet treatments, bees were starved for 4 h.

After 4 hours of starvation, bees were fed with carbohydrate, protein, and lipid sources for the diets. The normal diet included 50% (*w*/*v*) sucrose, 2% (*w*/*v*) casein amino acid and 1% (*v*/*v*) soybean oil (NDS) or palm oil (NDP), which was designed based on published literature and beekeeping practices to supply a spartan diet without substantial detrimental effects [50,51]. The high-fat diets contained an increased lipid level to 5% soybean oil (HFDS) or palm oil (HFDP). During this study, the diets were checked every 2 days to make sure each colony had an adequate dietary supply. Each bee was immobilized at 4 °C, and the whole-body wet weight of each bee was measured with an electric balance sensitive to 0.001 g. Following 7 days of these diet treatments, each bee was immobilized at 4 °C, the whole-body wet weight of each bee was measured with an electric balance sensitive to 0.001 g, whole guts and lipid bodies were dissected using fine-tipped forceps, blood hemolymph was collected, and RNA was extracted.

### 4.2. Determination of Honey Bee Hemolymph Glucose and Trehalose

Briefly, the hemolymph of each bee was removed by making a small incision at the neck and collected using a 10 µL pipette. Hemolymph was pooled from two to five honey bees to obtain 2–5 µL for assay, and specific care was given in order to avoid any contamination of the hemolymph from intestines. The concentrations of glucose and fructose were identified using enzymatic assay kit K-SUFRG (Megazyme, Bray Co., Wicklow, Ireland) as per the manufacturer’s instructions. The level of trehalose was measured using enzymatic assay reagent K-TREH (Megazyme, Bray Co., Wicklow, Ireland), with a ten-fold dilution because trehalose levels are higher than those of glucose. Glucose and trehalose standards were treated and used to quantify the sugar levels in hemolymph.

### 4.3. Determination of Honey Bee Body Lipid Content and Composition

Guts of 10–15 MF or CV bees were dissected into individual compartments using fine-tipped forceps and then were weighed. Lipid from the whole body (except gut) was extracted in chloroform/methanol with a chloroform/methanol method as previously described [52]. Prior to lipid extraction, the bee body was disrupted by homogenization in the presence of liquid nitrogen. Neutral lipids were separated on Silica gel 60 TLC plates (Merck, Darmstadt, Germany) and detected by charring. For quantification, individual lipids on the TLC plate, after visualization with iodine vapor, were recovered and transesterified to fatty acid methyl esters (FAMEs) using 1% sulfuric acid in methanol [53]. FAMEs were analyzed by GC-MS using the Agilent 7890 capillary gas chromatograph equipped with a 5975 C mass spectrometry detector and an HP-88 capillary column (60 m × 0.25 mm) (Agilent Technologies, Santa Clara, CA, USA) using a split ratio of 19:1. In addition, C17:0 was used as the internal standard for quantification.

The honey bee fat body is the principal site of stored fat and functions as both adipose and liver, thus lipid droplets in fat body were stained using hematoxylin and eosin to check the difference between normal diets and high-fat diets. Control diet contains 50% (*w*/*v*) sucrose, 2% (*w*/*v*) casein amino acid and 1% (*v*/*v*) lipid mixture for insect cell culture (Sigma-Aldrich, St Louis, MO, USA). Fat body tissue was dissected into PBS from the floor and roof of the abdomen, as described [54], and visualized under a microscope (Eclipse, Nikon, Japan).

### 4.4. RNA Extraction and Transcriptome Sequencing

Total RNA was extracted from individual samples using the Quick-RNA MiniPrep kit (Zymo Research, Irvine, CA, USA) according to the manufacturer’s protocol. RNA integrity was qualified using the RNA Nano 6000 Assay Kit of the Bioanalyzer 2100 system (Agilent Technologies, CA, USA). RNA sequencing libraries were generated using NEBNext Ultra RNA Library Prep Kit for Illumina (New England BioLabs, Ipswich, USA) and index codes were added to attribute sequences to each sample. Clusters were generated using TruSeq PE Cluster Kit v3-cBit-HS (Illumina, San Diego, CA, USA), and RNA-seq libraries were sequenced on an Illumina NovaSeq platform. 150bp paired-end reads were generated. The sequencing quality of individual samples was assessed using BWA with default parameters [55]. Gene expression was quantified using HTSeq v0.7.2 (https://doi.org/10.1093/bioinformatics/btu638). Differential expression analysis was performed using the DESeq2 package (v1.20.0) [56]. Gene ontology (GO) enrichment analysis was implemented by the clusterProfile R package. Functional analysis of DEGs was performed based on KEGG Orthologue markers using clusterProfiler v3.10.1 (https://doi.org/10.1089/omi.2011.0118). Raw sequence reads have been deposited at NCBI SRA database under the BioProject accession number PRJNA681941.

### 4.5. Gut Microbiota Analysis by High-Throughput 16S rRNA Sequencing

Each dissected whole gut was homogenized with a plastic pestle and total genome DNA was extracted from individual samples using a cetyltrimethylammonium bromide (CTAB) bead-beating method. Dissected gut was re-suspended in 728 µL CTAB, 2 µL β-mercaptoethanol, and 20 µL of 20 mg/mL proteinase K (Sigma-Aldrich, St Louis, MO, USA). Then the samples were transferred to bead-beating tubes containing 0.1 mL, 0.5 mm and 0.4 mL, 0.1mm silica zirconia beads (BioSpec Products, Bartlesville, OK, USA), and these tubes were processed in the bead beater for 2 min, placed on ice for 1 min, and bead-beaten for 2 min. Samples were incubated at 56 °C overnight. Then, 5 µL of 10 mg/mL RNase (Sigma-Aldrich, St Louis, MO, USA) was added to each sample, which were vortexed briefly and placed at 37 °C for 1 h. Finally, the DNA was extracted with phenol-chloroform-isoamyl alcohol (25:24:1), centrifuged, and the aqueous phase was alcohol precipitated, washed, and air-dried prior to resuspension in 50 µL nuclease-free water.

16S rRNA genes of V3–V4 regions were amplified using specific primers (515F: 5′-GTGCCAGCMGCCGCGGTAA-3′ and 806R: 5′-GGACTACHVGGGTWTCTAAT-3′). All PCR reactions were carried out with 15 µL of Phusion® High-Fidelity PCR Master Mix (New England Biolabs, Ipswich, USA), with 0.2 μM each of forward and reverse primer, and 10 ng template DNA. Thermal cycling consisted of one cycle of 1 min at 98 °C, thirty cycles of 10 s at 98 °C, 30 s at 50 °C, 30 s at 72 °C, and a final cycle of 5 min at 72 °C. PCR products were detected by electrophoresis and purified with Qiagen Gel Extraction Kit (Qiagen, Hilden, Germany). High-throughput sequencing was performed at Novogene Bioinformatics Technology Co. Ltd., Beijing, China on an Illumina NovaSeq6000 platform. Sequencing libraries were prepared with TruSeq® DNA PCR-Free Sample Preparation Kit (Illumina, San Diego, CA, USA) following manufacturer’s instructions and 250 bp paired-end reads were generated.

The quality of the raw data, as well as the number of sequences and operational taxonomic units (OTUs) were evaluated the QIIME (Version 1.9.1) [57]. The α-diversity (the Shannon estimator) and PCoA based on unweighted Unifrac Bray-Curtis distance were visualized in RStudio (RStudio: Integrated Development for R; RStudio, Inc., MA, USA). In univariate analysis of gut microbiota and predicted KEGG biochemical pathways in each group, a paired t-test or a Wilcoxon matched-pairs test was adopted. Beta diversity was calculated and visualized by generating principal coordinate plots. Functional inference of the bacterial community was made by PICRUSt analysis of the OTUs obtained from the Greengenes reference database [58]. Raw sequence reads have been deposited at NCBI SRA database under the BioProject accession number PRJNA681151.

## 5. Conclusions

Dietary-fat-induced obesity is associated with an increased risk for metabolic disorders, including diabetes, cancer, and heart disease. The polygenic and multiorgan nature of dietary-fat-induced obesity makes it difficult to discover the relative contributions to the disease. In order to examine these complicated crosstalks in a simple system, we established a dietary-fat-induced obesity model in the honey bee to elucidate the underlying mechanisms. We explored palm oil and soybean oil, two regular edible oils, as candidate dietary fats for influences on host metabolic status. Our results in honey bee support the view that HFD resulted in increased body weight gain and circulating glucose/trehalose levels, and accumulation of adipose, as in other vertebrate and invertebrate models. However, these detrimental effects differ between palm and soybean oils, which is associated with the dietary ratio of SFA/UFA. Transcriptomics showed that the lipid metabolism and immune-related pathways were remarkably regulated by different types of dietary fat. Moreover, the gut community structure and the microbial functions were altered by HFD of the two oils. Our results reveal the divergent effects of dietary fat with different compositions of fatty acids, and strengthen the use of the honey bee as a model organism to examine the crosstalk between dietary fats and gut microbiota implicated in host homeostasis.

## Figures and Tables

**Figure 1 ijms-22-00834-f001:**
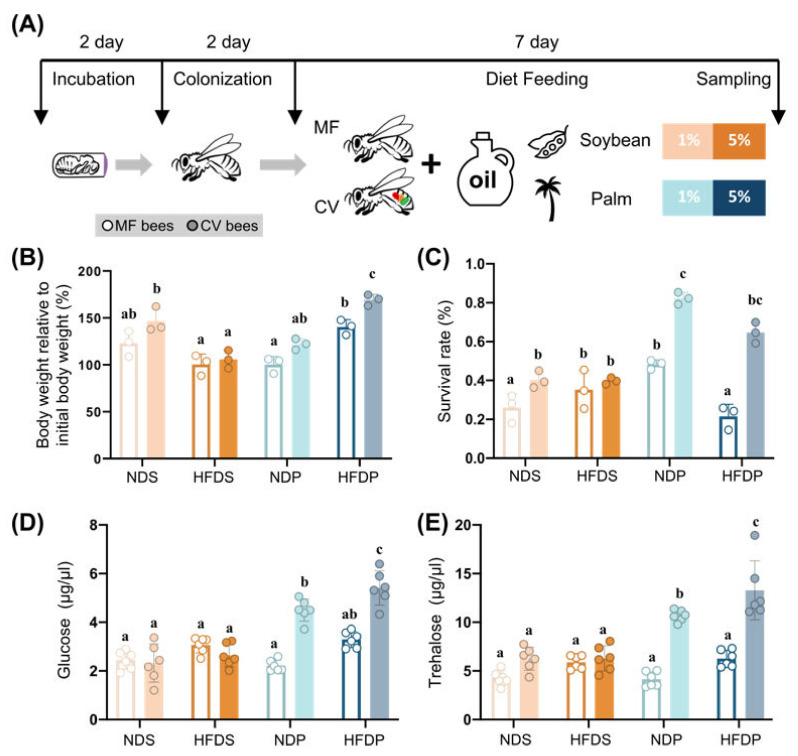
HFD (high-fat diet) impacts metabolism of honey bees with or without a gut microbiota. (**A**) Scheme of experiment. MF (microbiota-free) and CV (conventional gut microbiota) bees were generated from pupae emerged in the lab. They were fed diets containing 1% or 5% of fats from palm or soybean oil for 7 days. (**B**) Body weight gain as a percentage of initial body weight for both MF and CV bees on different amounts and types of dietary fat. Each dot represents the average value of bee individuals (*n* = 25) from one cup cage. (**C**) The survival rate of bees on different groups of diets for 7 days. Each dot represents the average value of bee individuals (*n* = 25) from one cup cage. (**D**) Glucose concentration in the hemolymph of individual bees (*n* = 6) from different diet groups. (**E**) Trehalose concentration in the hemolymph of individual bees (*n* = 6). Significant differences between groups were determined by one-way ANOVA with Tukey’s multiple comparisons test.

**Figure 2 ijms-22-00834-f002:**
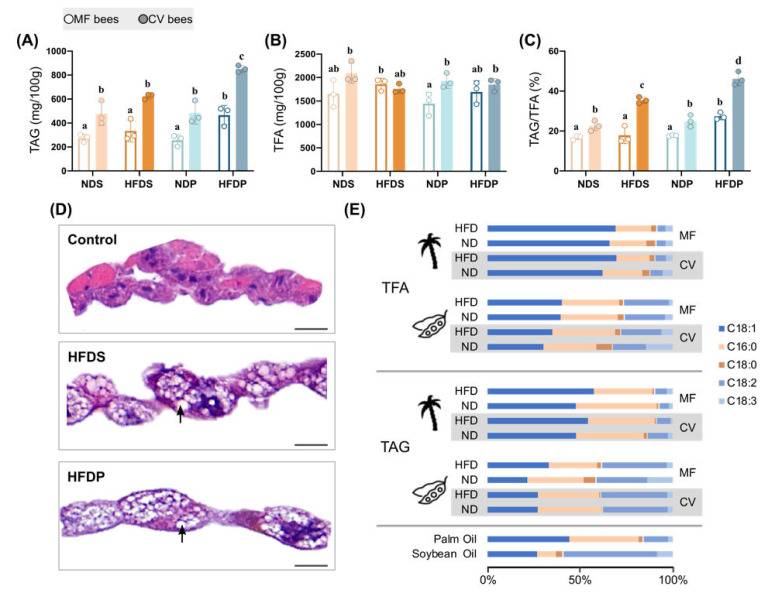
HFD-fed honey bees caused adipose fat accumulation. (**A**) TAG (triglycerides) concentration of the whole body of MF (*n*= 3) and CV (*n* = 3) honey bees from different diet groups. (**B**) Total TFA (total fatty acid) levels of the whole body of honey bee individuals (*n* = 3). (**C**) The ratio of TAG/TFA. (**D**) HE staining of control and HFD-fed bee fat bodies. Lipid droplets are marked using black arrows. The scale bar (bottom right) represents 10 µm. (**E**) Composition of SFAs (saturated fatty acids; C16:0, C18:0) and UFAs (unsaturated fatty acids; C18:1, C18:2, C18:3) in TAG and TFA of honey bees. The fatty acid composition of the soybean and palm oil used here was also quantified. Significant differences between dietary groups were determined by one-way ANOVA with Tukey’s multiple comparisons test.

**Figure 3 ijms-22-00834-f003:**
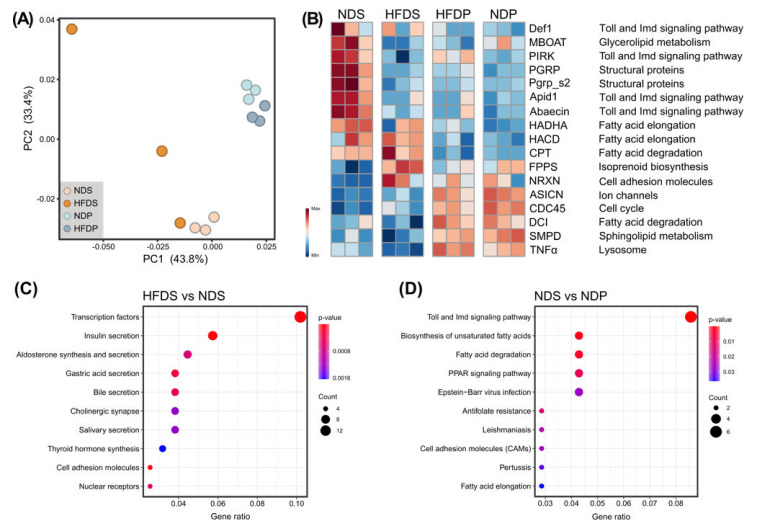
Transcript-level expression analysis of the gut epithelial of honey bees from different dietary groups. (**A**) PCoA (principal coordinate analysis) of the RNA-seq data from the gut epithelial cells of honey bees (*n* = 3) collected from different dietary groups. (**B**) Heatmap displaying DEGs (differentially expressed genes) among four different experimental groups, showing the expression profiles of each dietary fat cluster as indicated on top. The expression of genes are colored from blue to red according to the expression level from low to high. Each column represents one bee individual sample. (**C**) Representative enriched KEGG pathways on a HFDS (high-fat diet, soybean oil), compared to NDS (normal diet, soybean oil). (**D**) Representative enriched KEGG pathways on a NDS, compared to NDP (normal diet, palm oil).

**Figure 4 ijms-22-00834-f004:**
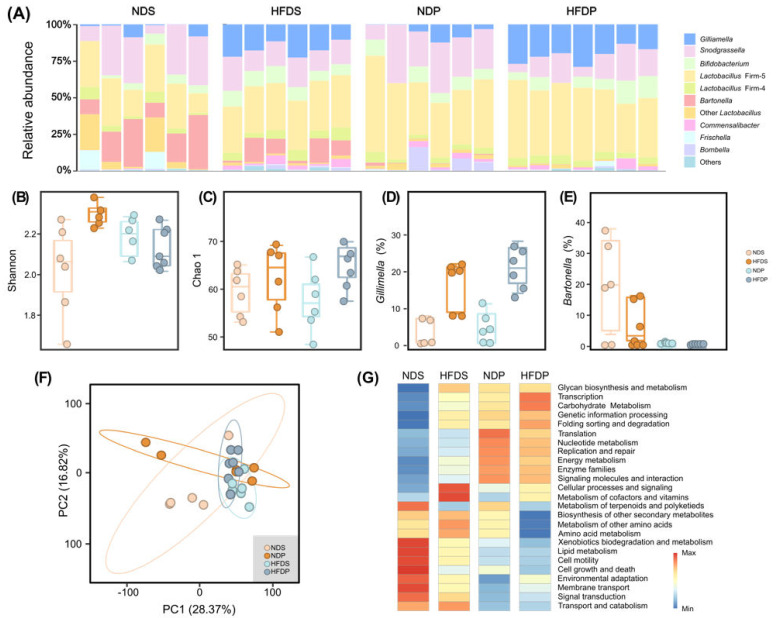
Changes in the composition and functional potential of the bee gut microbiota after ND (normal diet) and HFD intervention in each group. (**A**) Bar graph of bacterial abundance at the genus level. Genomic DNA was extracted from the gut samples taken from honey bee maintained for 7 days on different dietary diets. Samples were analyzed for the bacterial composition by sequencing of the bacterial 16S rRNA fragments (*n* = 6). (**B**) α-Diversity at genus level estimated by Chao1 richness index. (**C**) Shannon diversity estimator. (**D**) The relative abundance of *Gilliamella* in the bee gut. (**E**) The relative abundance of *Bartonella* in the bee gut. (**F**) PCA plot describing functional inferences (PICRUSt) of bacterial communities across dietary fat treatments. (**G**) Heatmap displaying the differentially enriched KEGG pathways (Level 2) prediction by PICRUSt across dietary groups.

## Data Availability

Data available in a publicly accessible repository.

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
