# Peer review of "High-Fat Diets with Differential Fatty Acids Induce Obesity and Perturb Gut Microbiota in Honey Bee"

_ijms, 2021, doi:10.3390/ijms22020834_

Round 1

Reviewer 1 Report

This is a sound experimental work and a well written MS.

The authors tried control and 5% of oil levels feeding honeybees.

The 10% of oil level was found to be toxic.

I would like to ask the authors to justify further in the discussion of the paper why they tried only two levels of oils and how they decided to choose 5% oil as the high oil diet. At the moment, the rationale of this choice is not clear. Also, in the discussion, it would be useful to have a paragraph on the value of their data and extrapolate them to human consumption of oils.

Reviewer 2 Report

Dear authors, your study entitled "High-fat diets with differentially fatty acids induce obesity and perturb gut microbiota in honeybee" is scientifically sound and very well written. I also like the overall presentation including the figures. It is touching upon a very hot and interesting topic: the use of honey bee for the functional study of gut microbiome.

Overall this paper could be published as such. I only have a coupe of small remarks

-Page3-Line5: there is a typo. Honeybees (instead of "hoeybees)

Sometimes is honey bee written in 2 words. Just doublecheck throughout the plain text please.

-Figure 1&Figure 2: the letter "a, b and c" should be mentioned in the caption.
